# The Dunn Ranch Academy: Developing Wildland Fire Literacy through Hands-On Experience with Prescribed Fire Science and Management

**Devan Allen McGranahan** [1,*,†] **, Craig Maier** [2] **, Ryan Gauger** [3] **, Chris Woodson** [4,‡] **and Carissa L. Wonkka** [5,†]

1   Livestock & Range Research Laboratory, USDA Agricultural Research Service, Miles City, MT 59301, USA
2   Tallgrass Prairie and Oak Savanna Fire Science Consortium, University of Wisconsin, Madison, WI 53715, USA; cmaier2@wisc.edu
3   Missouri Field Office, The Nature Conservancy, St. Louis, MO 63144, USA; rgauger@tnc.org
4   Big Muddy National Fish and Wildlife Refuge, US Fish and Wildlife Service, Booneville, MO 65233, USA; chris_woodson@fws.gov
5   Northern Plains Agricultural Research Laboratory, USDA Agricultural Research Service, Sidney, MT 59270, USA; Carissa.Wonkka@usda.gov
\*   Correspondence: Devan.McGranahan@usda.gov
†   The US Department of Agriculture is an equal opportunity lender, provider, and employer.
‡   The findings and conclusions in this article are those of the authors and do not necessarily represent the views of the US Fish and Wildlife Service.

**Abstract:** Wildland fire literacy is the capacity for wildland fire professionals to understand and communicate fundamentals of fuel and fire behavior within the socio-ecological elements of the fire regime. While wildland fire literacy is best developed through education, training, and experience in wildland fire science and management, too often, development among early-career professionals is deficient in one or more aspects of full literacy. We report on a hands-on prescribed fire methods workshop designed to provide training and experience in measuring and conducting prescribed fire, with a focus on grassland ecosystems. The workshop was held in March 2022 at The Nature Conservancy's Dunn Ranch Prairie in northern Missouri. It consisted of hands-on training and experience in measuring fuels, fire weather, and fire behavior. Prescribed fire operations training facilitated both hands-on learning and vicarious learning by rotating squad roles among several small sub-units on the first day of live fire exercises. Participants then gained experience as crew members for two larger prescribed burns (60 and 200 ha). We report here on the successes and lessons learned from the perspectives of both participants and the instructor cadre for what was widely regarded as a successful workshop.

**Keywords:** #NerdTREX; #FireScienceDIY; learn and burn; wildland fire science literacy

## 1. Introduction

Rapidly addressing and effectively managing the "wicked problems" of wildland fire management–from controlling wildfires to administering prescribed burns in fire-dependent ecosystems–demands *wildland fire literacy*–the capacity for wildland fire professionals to understand and communicate fundamentals of fuel and fire behavior within the socio-ecological elements of the fire regime (Figure 1).

However, existing modes of education, training, and experience-building are slow to bridge persistent gaps between wildland fire science and management. In this Feature Paper, we summarize the history and state of professional development in the wildland fire community and describe the motivations, successes, and opportunities to improve a novel mode of integrating both training and experience in both studying and conducting a prescribed fire.

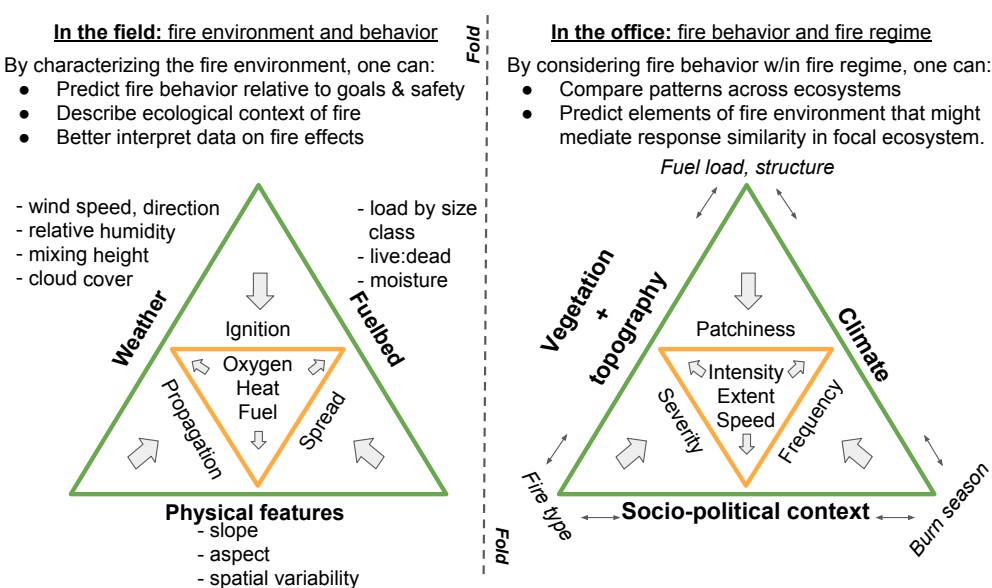

**Figure 1.** A wallet card describes the components of the fire environment most relevant to two arenas of the wildland fire professional: In the field, and in the office [1]. Terms related to the socio-ecological context of the fire regime follow McGranahan and Wonkka [2].

## 1.1. Background on Wildland Fire Management and Training

> In the measurement of fire weather and forest inflammability . . . it is necessary to use many methods peculiar to this work. Some of these methods are familiar to meteorologists, but few foresters have had any appreciable training in meteorology. Others are of such recent development and so specially designed for forest protection that they are unknown to most meteorologists and are not yet taught in the schools of forestry or described in any textbooks. (H. Gisborne [3], p. 1).

While the United States wildland fire community has long recognized the need to improve education and training in fuels, fire behavior, and management, a cohesive curriculum encompassing the science and practice of wildland fire use has yet to emerge. On one hand, some of the original standards for fire management and research on the fire environment were developed in tandem, given the applied emphasis Harry Gisborne placed on using his seminal work on weather and fuel moisture to inform the preparation, deployment, and safety of fire control resources, beginning in the 1920s [4]. On the other hand, while theoretically applicable to using wildland fire as well as fighting it, these standards were solidly oriented within the mode of fire suppression: *the sole purpose of the weather and inflammability measurements described herein is to improve forest-fire control* [3]. The US Forest Service, having effectively defeated any support of "light burning" or other wildland fire use, controlled fire research funding as early as 1928 and by 1935 clearly established an aggressive policy of suppression [5]. Enclaves of advocates for fire use moved their discussions beyond the reach of the anti-fire establishment, such as the Tall Timbers Fire Ecology Conference [5].

Management agencies began to adopt prescribed burning through the 20th century. Legislative, bureaucratic, and even cultural changes first opened the National Park Service then the US Forest Service to prescribed fire that included both pre-planned and intentionally-set burns and natural ignitions allowed to spread through designated wilderness areas under prescribed conditions [5]. The U.S. Fish and Wildlife Service (USFWS) ignited its first prescribed fire in 1927 on St. Marks National Wildlife Refuge in Florida. State conservation agencies began conducting prescribed fires around this time as well; for instance, the Wisconsin Department of Conservation used prescribed fire for the first time in 1939 [6]. The Nature Conservancy conducted its first prescribed fire in 1962 (Figure 2; refs. [5,7]).

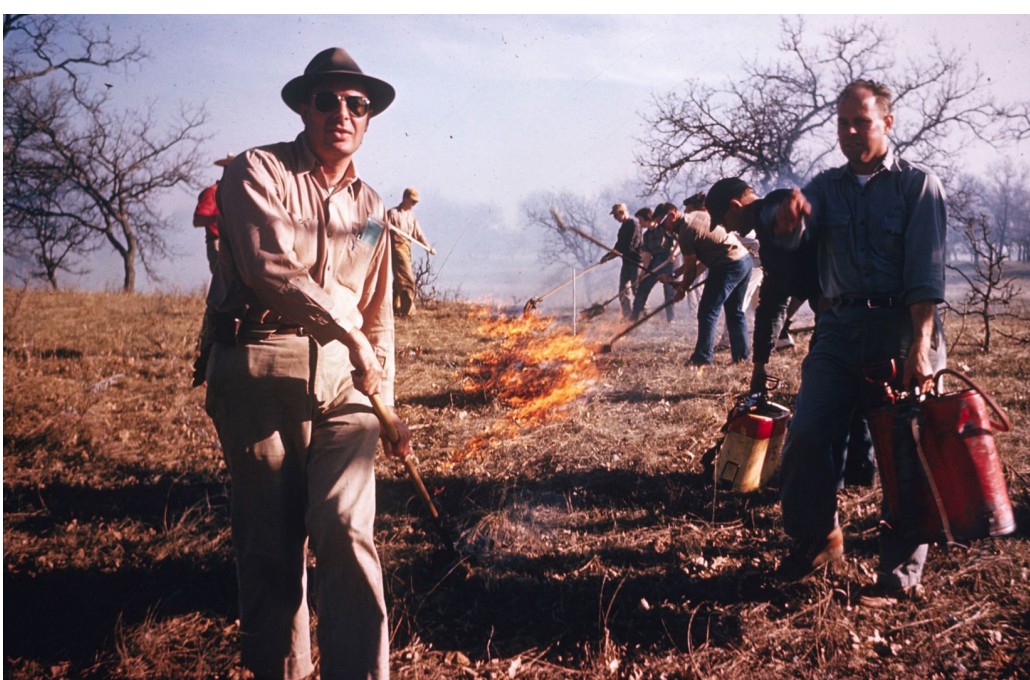

**Figure 2.** The Nature Conservancy conducted its first prescribed fire at the Helen Allison Scientific and Natural Area, Minnesota, in 1962.

Meanwhile, the management of wildland fire operations developed as well, becoming more specialized as standardized command-and-control systems evolved. Disastrous fires in 1970 prompted developments that became the National Wildfire Coordinating Group (NWCG) and the Incident Command System (ICS), which the NWCG in turn adopted and agencies had widely employed by 1985 [8,9]. The ICS has facilitated cross-boundary collaboration among agencies and jurisdictions in the US as well as among participating nations [10]. However, some local fire departments bristle at the constraints of the ICS [11], and the hierarchical structure of training, certification, and qualification for positions within the ICS can create a barrier to allowing otherwise experienced personnel to conduct prescribed burns [12]. In fact, as early as the 1980s, the USFWS condensed the content of 140 hours worth of NWCG coursework into a 36-hr course that combined principles of both prescribed fire and wildfire suppression, to reflect the changing demands on USFWS personnel [13].

*1.2. State of Wildland Fire Education and Training Today*

The ideal system for preparing the next generation of fire professionals would integrate and/or provide education, training, and experience in parallel. Such a system would share characteristics with educational models used in other professions such as law, business, and medicine, where coursework is offered in conjunction with summer job experiences, training courses, and extensive internships. (Kobziar et al. [14], p. 344).

Unfortunately, wildland fire professionals, particularly those in seasonal or collateral duty roles or those where fire is only a single component in a longer list of duties, rarely achieve a sufficient amount of training, education, and experience. This means that many of those in collateral fire positions may never achieve sufficient training, education or experience, and if they do it is often a long, arduous, and not always straightforward process. Kobziar et al. [14] identified three common syndromes of lopsided professional development (Figure 3). The main issues are disparities between *education* (receiving knowledge on the fire environment, fire effects, and how and why one might conduct a prescribed fire), *training* (being taught how to use and apply various fire management resources), and *experience* (a background of having performed fire management tasks).

Employees of agencies and non-governmental organizations with substantial training and experience, such as seasonal firefighters and other crew members, have often received a limited education in fire science and encounter limited opportunities to receive more instruction. Meanwhile, graduates of academic programs, academic researchers, and staff in professional positions might have a substantial amount of education and even some training, but lack experience working on actual fires. For example, technical specialists such as Resource Advisors (READ) need only to meet basic Firefighter Type 2 (FFT2) certification and physical fitness requirements before being assigned to complex incidents: *the bottom line is that, in my opinion, the general lack of hands-on training of fire archaeologists after they earn their red cards is a recipe for disaster* ([15], p. 3).

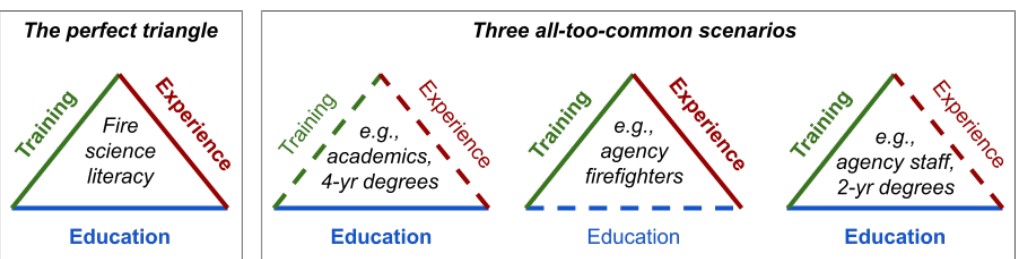

**Figure 3.** Kobziar et al. [14] identified three components of professional development in the wildland fire community that we identified as essential to developing wildland fire literacy. Unfortunately, all too often, certain syndromes of lopsided professional development persist among various members of the wildland fire community.

Gaining experience with prescribed fire, specifically, has been a persistent problem. Writing in 1985, Heitlinger and Davis [16] highlighted the need for hands-on experience with prescribed fire in their review of available workshops and university courses related to fire ecology and management. Decades later, just a few universities offer programs in prescribed fire, specifically, and only a subset of those provide hands-on experience with live fire [14]. Oklahoma State University is one example that augments classroom-based courses in fire ecology with experiential learning that includes conducting prescribed fires; a survey of participants indicated that hands-on experience with prescribed fires was the most valuable opportunity of the program [17]. Outside of the university setting, opportunities for hands-on experience with prescribed fire range from demonstrations aimed at landowners (e.g., Society for Range Management's Range Practicum; [18]) to multi-day learn-and-burn operations (e.g., Prescribed Fire Training and Exchanges, or TREX; [12]).

### 1.3. Bridging Gaps between Science and Management

There are reasons other than to smell smoke and escape from the classroom to justify the time and expense involved in setting up a fire exercise. ([19], p. 50)

Despite these advancements in prescribed fire experience, there remains a paucity of fire science integration in the education and training of wildland fire and natural resource professionals. In many tertiary natural resource programs, education is often limited to fire effects on natural resources, with students gaining little exposure to interactions between fuels, weather, and the fire behavior that drives fire effects. As for training, only at the highest levels of leadership or position specialization do managers pursuing professional development through the ICS receive extensive and comprehensive training in the wildland fire environment (Table A1). Basic wildland firefighter certification includes only a minimal amount of training in fire behavior (S-190: 7 h in-person and 6–8 h online). The second fire behavior course, S-290, introduces interactions between fuels and topography, weather, and fire behavior, but is primarily aimed at training supervisors to recognize potentially dangerous conditions for their crew. Even within the context of wildfire suppression, training to recognize and mitigate hazards associated with extreme fire behavior does

not reflect the most recent scientific understanding, and the needs of crew leadership on the fireline often differ from those of incident command [20]. Knowledge gaps between research and prescribed fire management appear to be even less recognized in the wildland fire community—review of barriers to integrating science in wildland fire management found literature focused primarily on "wildfire management" and secondarily on "fuels management", with no specific mention of prescribed burning [21].

Although gaps between researchers and practitioners have been described in almost every professional field from health care to conservation biology [22,23] and across land management broadly [24], gaps between wildland fire scientists and managers can be particularly wide. Often, the gaps can be literal distance in space and time, in the sense that many scientists cannot participate in or even directly observe fire management operations that adhere to ICS requirements for training and certification. ICS protocols have expanded from federally-managed wildfire incidents to prescribed fire operations managed by state agencies and NGOs—for example, TNC adopted NWCG ICS standards in the early 2000s. As such, scientists are increasingly distanced from making real-time observations and measurements of fire as it happens.

Not only are education, training, and experience necessary to develop wildland fire literacy, the triad must be developed in both the realms of fire science and management While completion of online coursework can help clear some administrative hurdles for scientists posed by ICS requirements, there are few substitutes for the experience gained by experiencing a live fire. Observation of live fire has long been recognized as a critical factor in understanding fire behavior and the challenges it poses to fire management [19,25]. Understanding how managers conduct safe prescribed burns and the various constraints (time, weather, policy) that managers must weigh against desired fire behavior is essential to designing feasible, and effective, fire science research protocols. Conversely, understanding which components of the fire environment can be measured—and how best to do so—ought to help fire managers incorporate new fire science information into their planning and operations.

Here, we report on a Hands-on Fire Science Methods workshop designed to promote wildland fire literacy, held in the Midwestern US in the spring of 2022. We provide an overview of the workshop's objectives and activities, as well as a reflective critique in the form of "lessons learned" informed by a group debriefing of the workshop leadership and instructors (the cadre) and an anonymous online survey of workshop participants. While the workshop was widely viewed as a success, we also discuss elements that merit consideration or improvement in future iterations of this workshop or others with similar objectives and/or audiences.

## 2. Briefing

Here, we describe the intentions and operations of the course.

### 2.1. Leaders' Intent

The workshop was intended to provide early-career fire professionals hands-on experience with tools and techniques relevant to prescribed fire science and management, with a focus on grassland ecosystems. The objective was to develop wildland fire literacy by emphasizing two distinct arenas: Best practices for conducting robust wildland fire science, including collecting data on fuels, fire weather, and fire behavior; and strategies and tactics for safe and effective prescribed fire operations. Broadly speaking, the workshop was designed to meet the objective by providing equal opportunity for early-career professionals to learn and experience both fire science data collection and conducting prescribed burns, regardless of their familiarity–or lack thereof–with either arena. Activities were designed to be as hands-on as possible and aimed to provide ample opportunity for students to learn from course instructors as well as other participants.

A secondary intention was to provide career development opportunities for participants with specific needs that fit into the workshop. This was aimed mostly at trainees with

open *Position Taskbooks* (https://www.nwcg.gov/publications/position-taskbooks/about; accessed on 15 August 2022) for qualification to achieve ICS positions. While no such opportunities were guaranteed to participants, leadership recognized that several participants had open taskbooks for which members of the teaching cadre could provide signatures if the opportunity for trainees to perform tasks arose. As such, the workshop was run as an incident just like any other prescribed fire or wildfire response. Prescribed Fire Training Exchanges (TREX) are structurally similar, with a core group of instructors (cadre) and each participant serving as both a trainee as well a trainer depending on their individual skill set and qualifications.

*2.2. Procedure*

Here, we describe logistical considerations from the perspective of planning a successful operation that included as much flexibility as possible to satisfy workshop objectives without letting too much chaos show through the cracks.

2.2.1. Date and Location

The Hands-on Fire Science Methods Workshop was held 14–18 March 2022 at The Nature Conservancy's Dunn Ranch Prairie near Eagleville, Missouri, USA (Figure 4). This week was specifically selected from a review of spring break schedules for popular natural resource universities in the Midwestern US to accommodate as many graduate students as possible. While several locations within the Midwest were considered, TNC's Dunn Ranch Prairie met several important criteria:

- *Large area of potential burn units*. Dunn Ranch/Pawnee Prairie is a complex of nearly 1620 ha (4000 acres) with burn units that range from 16 to 200 ha (40–500 acres), many of which could be broken into smaller units to accommodate training opportunities.
- *Latitude conducive to spring fire*. Although the weather is always an uncontrollable variable, mid-March in northern Missouri is typically warm and dry enough for prescribed burning, but spring has typically not progressed to the point that vegetation is overly green.
- *On-site facilities for accommodation and instruction*. Recent infrastructure improvements at the Dunn Ranch include a bunkhouse with shared kitchen and laundry facilities, in addition to indoor and outdoor common areas for group instruction and communal meals. Upstairs from participant quarters are accommodations for the cadre and facilities for their daily planning meetings.
- *Local and regional fire resources*. MO TNC has been rebuilding its fire program and coordinating efforts locally with conservation partners as well as coordinating with the IA TNC fire program. While Dunn Ranch staff does have a contingent of equipment and trained staff on-site, it is not enough to independently conduct fire operations. TNC MO is now engaging with cooperators in the state and region especially the Missouri Department of Conservation (MDC) and the USFWS to have greater capacity. MDC regional staff manage a portion of the Pawnee Prairie Natural Area (adjacent to Dunn Ranch) with prescribed fire and the USFWS has active prescribed burn programs on Loess Bluffs and Neal Smith National Wildlife Refuges (NWRs), both within 2 h of Dunn Ranch Prairie.

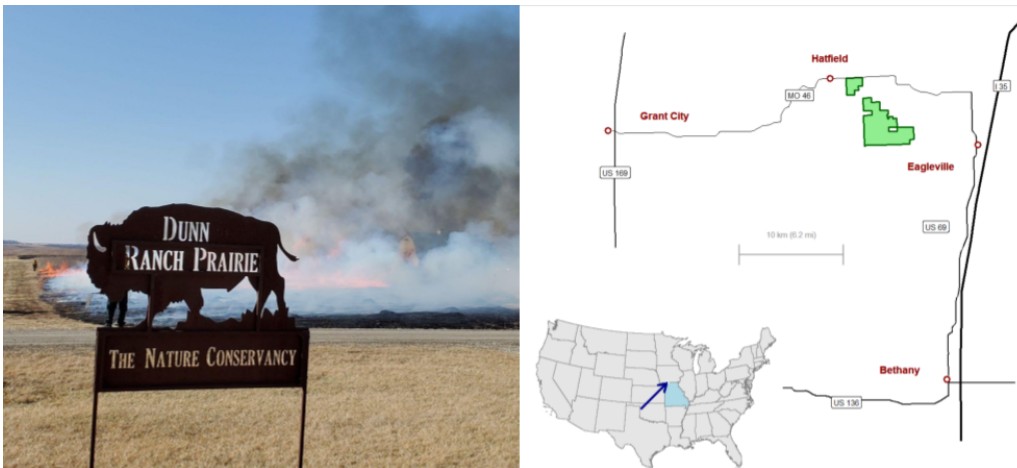

**Figure 4.** The Nature Conservancy's Dunn Ranch Prairie, in northern Missouri, USA, hosted the 2022 hands-on workshop in fire science methods. In the large map on right, the Dunn Ranch is highlighted in green and indicated on the inset by the blue arrow.

Food was provided to workshop participants. Basic breakfast and lunch provisions were available in the communal dining every day for self-constructed meals. Each evening, a local restaurant catered a hot buffet in the communal dining area.

2.2.2. Personnel and Equipment

Workshop leadership was divided among a cadre of fire science and fire management professionals, organized under a Prescribed Fire Burn Boss (RXB2) in charge of fire management planning and decision-making for TNC Missouri. The cadre met virtually several times prior to the workshop to coordinate roles, responsibilities, and logistics, and during the workshop met nightly to debrief and plan the next day's activities. All members of the cadre were, at a minimum, certified as FFT2 under the ICS and current in fitness tests to ensure their availability to contribute to all components of the workshop.

The USDA Agricultural Research Service provided two scientists to lead fire science modules; as each is experienced in conducting prescribed fire, the fire science instructors also supported fire operations modules by serving as additional fireline supervisors. In addition to TNC's burn boss, TNC and the US Fish and Wildlife Service provided personnel qualified as squad bosses (Firefighter Type 1—FFT1) and single resource bosses (Firing Boss—FIRB and Engine Boss—ENGB). Having led the planning and recruitment phases ahead of the workshop, the coordinator of the Tallgrass Prairie and Oak Savanna Fire Science Consortium assisted both fire science and operations modules as necessary in addition to handling logistics for participants on-site.

TNC and USFWS also provided all necessary equipment including hand tools, Type 6 and Type 7 engines, and all-terrain vehicles, as specified in TNC-approved burn plans. Local TNC resources at Dunn Ranch Prairie prepared burn units and provided communication, transport, and backup suppression resources.

Workshop enrollment consisted of 10 participants, split nearly equally between two broad groups: graduate students and early-career professionals in education, research, and outreach; and early-career professionals in natural resource management. Natural resource managers represented TNC, Quail Forever, and two tribal authorities in Minnesota.

On the first day, workshop participants were assigned to two, 5-person squads that remained consistent through the entirety of the workshop. Consistent squads addressed two goals: firstly, it simulates the close, interactive working environment that characterizes wildland fire operations and provides the opportunity for *crew cohesion*, which has been identified as a preventative factor in reducing accidents on incidents [26]. Secondly, this crew cohesion might also contribute to developing *bonds of empathy* among the group,

which has been associated with the emergence of intuitive thinking among group members in science education literature [27].

## 3. The Operation

Workshop participants were given educational materials at the beginning of the workshop that covered both fire science research methods and prescribed fire operations. The booklet served as the primary reference for all training modules and included multiple copies of datasheets and protocols for hands-on experience.

Each day of the workshop was divided into time for fire science or fire operations modules. The first day was dedicated to training and orientation with equipment for both Science and Operations modules, with introductions to application techniques for both module types, as well (Figure 5). The second and third days consisted of live fire exercises on TNC burn units (Fire Operations) in the afternoon, prior to which morning Fire Science modules consisted of measuring fuels and deploying instruments to collect fire weather and fire behavior data.

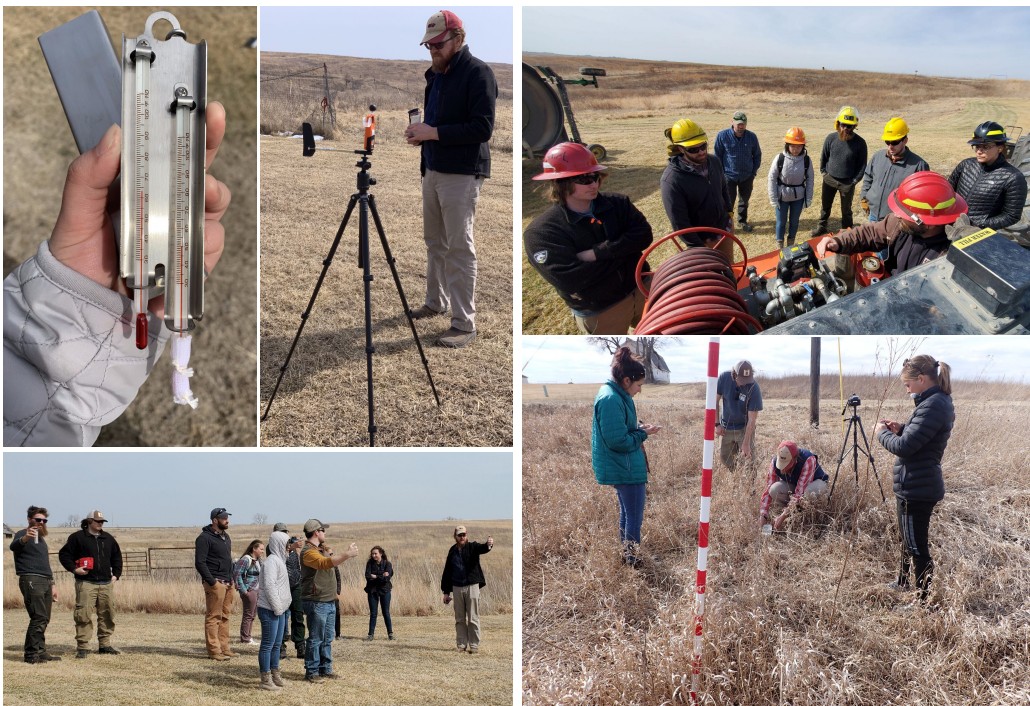

**Figure 5.** The first day of the workshop was focused on introducing students to the tools they would be using and training them on safe and effective operation ahead of live fire. Participants were introduced to measuring fire weather with both the Belt Weather Kit and automated weather stations, pump operations, and measuring fuel load with both destructive and non-destructive techniques.

### 3.1. Fire Operations

Lighting several fires is the only way to learn what environmental and fuel conditions are required to produce desired fire behavior. (McPherson et al. [28]).

Workshop participants were included in three live fire exercises. The first exercise was small-scale and focused on training and introducing students to the workshop chain of command, while the second and third exercises were focused on providing hands-on experience in studying and applying prescribed fire at the landscape scale.

#### 3.1.1. Training

The first live fire exercise was a series of seven small, independent burn units (sub-units) around the Dunn Ranch Prairie headquarters (Figure 6). Burns proceeded one at a time under the supervision of the TNC RXB2 burn boss. One qualified squad boss (FFT1)

and one fire science instructor was attached to each squad as mentors, available to answer questions and offer advice without distracting the burn boss from general oversight.

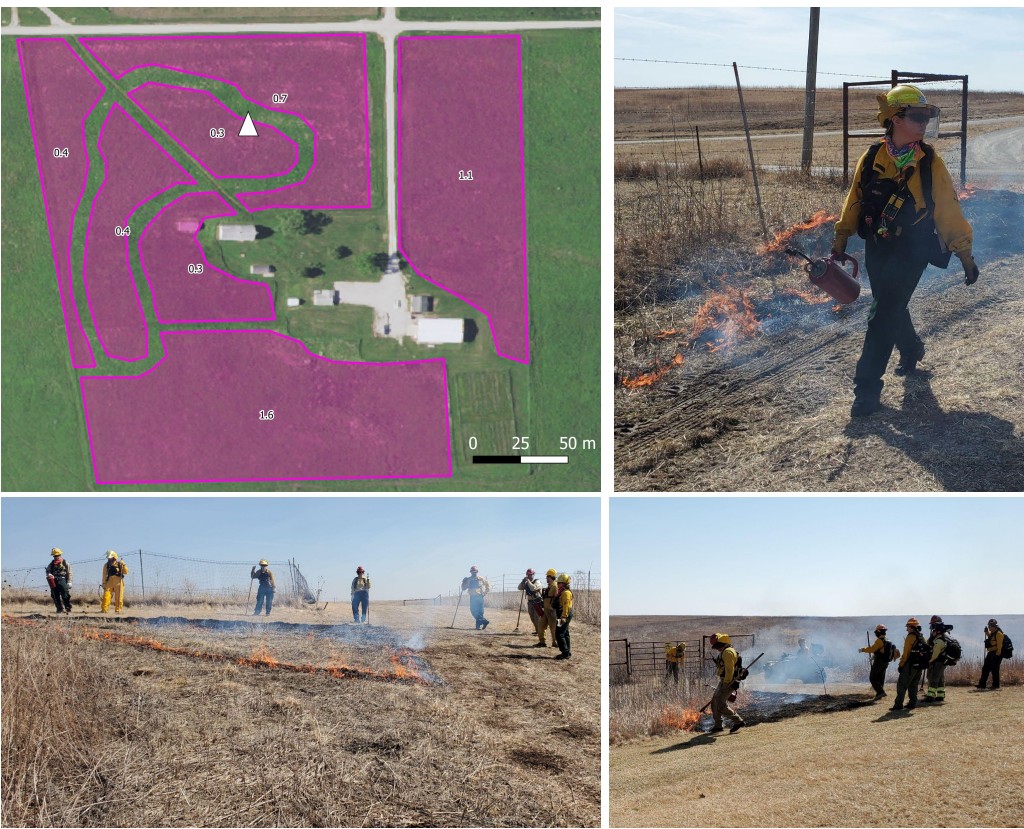

**Figure 6.** The seven sub-units around the Dunn Ranch Prairie headquarters were used for the training live fire exercises, in which squads swapped out as the burn crew among sub-units, and squad leadership rotated each time the squad was up to burn. Numbers denote hectares of each sub-unit and the white triangle indicates where the leadership allowed a spot fire to slop over the walking trail. Note that the perimeter of the headquarters area is maintained as lawn, and as such was short and green at the time of the workshop, serving as excellent firebreaks.

The burn for each sub-unit was planned and conducted by a single squad, with squads alternating among sub-units and squad leadership rotating among squad members each time the squad was up to burn. When not actively burning, the other squad was tasked with first debriefing their previous burn, scouting and planning their next burn, and watching their peers conduct the current burn. Both the on-off approach among squads and the rotation of leadership within squads facilitated two modes of experiential learning: *hands-on learning* and *vicarious learning*. There is evidence that students can learn as much, if not more, by observing their peers perform tasks (vicarious learning) than through their own learning by doing (hands-on learning) [29], although students often prefer the hands-on approach [30].

### 3.1.2. Experience

Participants received hands-on experience with prescribed fire by conducting two management-scale burns at Dunn Ranch Prairie (Figure 7). These units were approximately 60 and 200 ha (150 and 500 acres). Each live fire exercise began with a crew briefing on the management objectives of the burn, leadership structure and supervisory assignments, current and expected fire weather, available resources and personnel assignments, and contingency plans. In both cases, squads were assigned as crew members to one of two lines under the supervision of TNC and USFWS fire management personnel. Line officers rotated participants through tasks, emphasizing ignitions operations with drip torches;

holding operations with hand tools and pumps; and mop-up. Fire science instructors served primarily as lookouts and back-up crew members, and engaged participants in conversation about the operation or observed fire behavior as opportunity allowed.

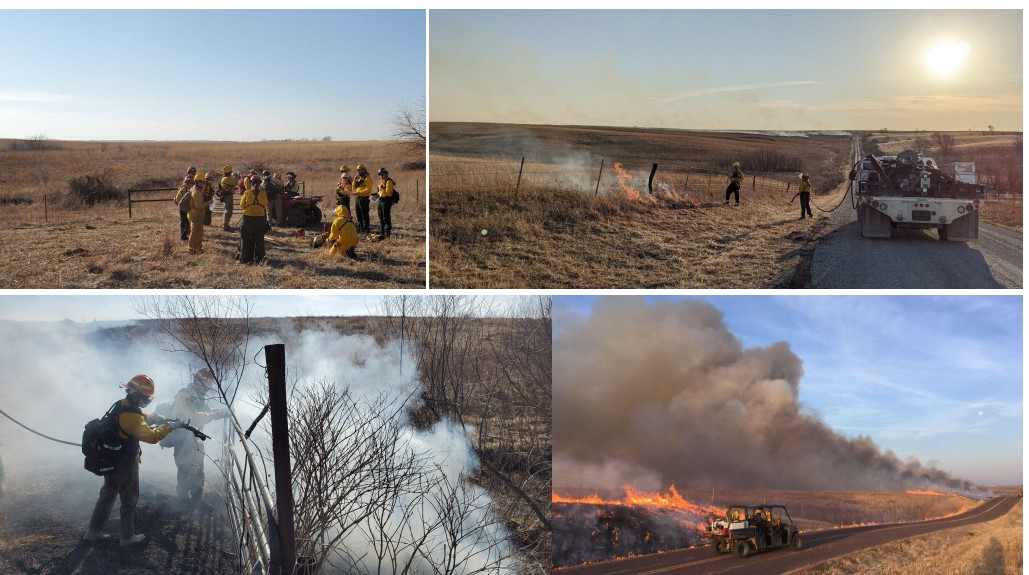

**Figure 7.** Workshop participants gained hands-on experience with prescribed fire as crew members burning two management units at Dunn Ranch Prairie. Clockwise from top left: The TNC RXB2 leads a briefing at the beginning of the first, 60-ha unit. Two course participants use a wet line from a TNC engine to burn out a fence line. Under supervision from a TNC line officer, a course participant uses water to limit the combustion of coarse woody debris in a ravine along the control line. Course participants patrol after setting a head fire from a county road along the 200 ha burn unit.

### 3.2. Fire Science

> Fire's ecology is not restricted to fire's "effects", but to the very properties that make open combustion possible. (S.J. Pyne [31], p. 126).

The second core element of the workshop was training and experience in field methods to quantify relevant components of the fire environment. Open combustion requires heat, oxygen, and fuel; how quickly and intensely fire spreads depends on the weather, vegetation, and several characteristics of the physical environment (Figure 1). Variability in fire effects can often be attributed to variability in how much energy organs, organisms, or soil particles are exposed to during heating, and variability in objective energy exposure at a given landscape position can be described by several key variables of fire weather and fire behavior [2].

### 3.2.1. Training

On the first day, workshop participants were introduced to tools and protocols for measuring fire weather, fuels, and fire behavior. For training in measuring fire weather, participants were first introduced to the standard belt weather kit, with a focus on measuring relative humidity with the sling psychrometer and wind speed with the *plastic venturi action wind meter* [32]. Participants were then introduced to an automated data logging weather station (Kestrel 5500FW). For both types of instruments, best practices for location and frequency of observations were discussed.

Fuel sampling consisted of both non-destructive and destructive sampling, using a Robel pole to classify visual obstruction readings (VOR; [33]), and clipping quadrats, respectively. Clipped quadrats were placed around the sight lines of the Robel pole to facilitate calibration of VOR with actual biomass [34]. Participants were also introduced to non-destructive measurements of total fuel load using a ceptometer, which compares photosynthetically-active radiation (PAR) above the plant canopy and at the soil surface to

estimate the amount of vegetation. Methods to measure live and dead fuel moisture and relative load were also described and tools such as a duff moisture meter for instantaneously measuring live fuel moisture demonstrated [35], but the live fuel component at the Dunn Ranch at the time of the workshop was negligible.

Participants were trained in measuring fire behavior with the FeatherFlame thermocouple datalogger system [36], which uses an open-source, Arduino-based microcontroller platform (Adafruit Industries; Brooklyn, NY, USA; adafruit.com, accessed on 15 August 2022) to read and log temperatures from industrial-grade K-type thermocouples (Omega Engineering; Norwalk, CT, USA; omega.com, accessed on 15 August 2022). Although temperature alone is often an inadequate measure to describe fire behavior in that it does not relate directly to an important driver of variability in fire effects–e.g., intensity, or the amount of energy released by combustion–temperature data from thermocouples are widely used in fire ecology [37].

A novel advantage of the FeatherFlame system over many conventional thermocouple datalogging systems is the simple integration of multiple thermocouple channels per datalogger. When arranged appropriately, such as in an equilateral triangle [38], simultaneous temperature records associated with a single timestamp facilitate measuring *two-dimensional rate of spread*. Many conventional measurements of fire spread rate–so-called 1-D measurements [39]–require direct observation along a pre-determined vector that is exactly perpendicular to an evenly-advancing flame front. However, large burn areas and complex ignition patterns often preclude direct observation, and uneven fuel or other obstacles that create variability in the flame front make a perpendicular observation vector difficult. However, the 2-D array can record the rate of spread without direct observation and is free of each of the above-mentioned pitfalls, thus translating data on temperature into more useful information on fire behavior.

### 3.2.2. Experience

Participants measured fuel load and deployed 2-dimensional fire behavior instrument arrays at three points within both units used in the large-scale live fire exercises. Squads rotated between fuels sampling and deploying thermocouple dataloggers. After receiving training directly from the fire science instructors, individuals within each squad then explained how the sampling schemes and systems worked to their peers (Figure 8).

The three flame temperatures recorded from the above-ground thermocouple array allowed calculation of the rate of spread for each fire, and thermocouple probes placed at the soil surface provided soil heating data (Figure 8). While there was considerable variability in flame temperatures within burns, soil surface heating was more consistent within and among burns, with the exception of a soil surface probe that might not have been correctly deployed on the first burn (15 March, Figure 8).

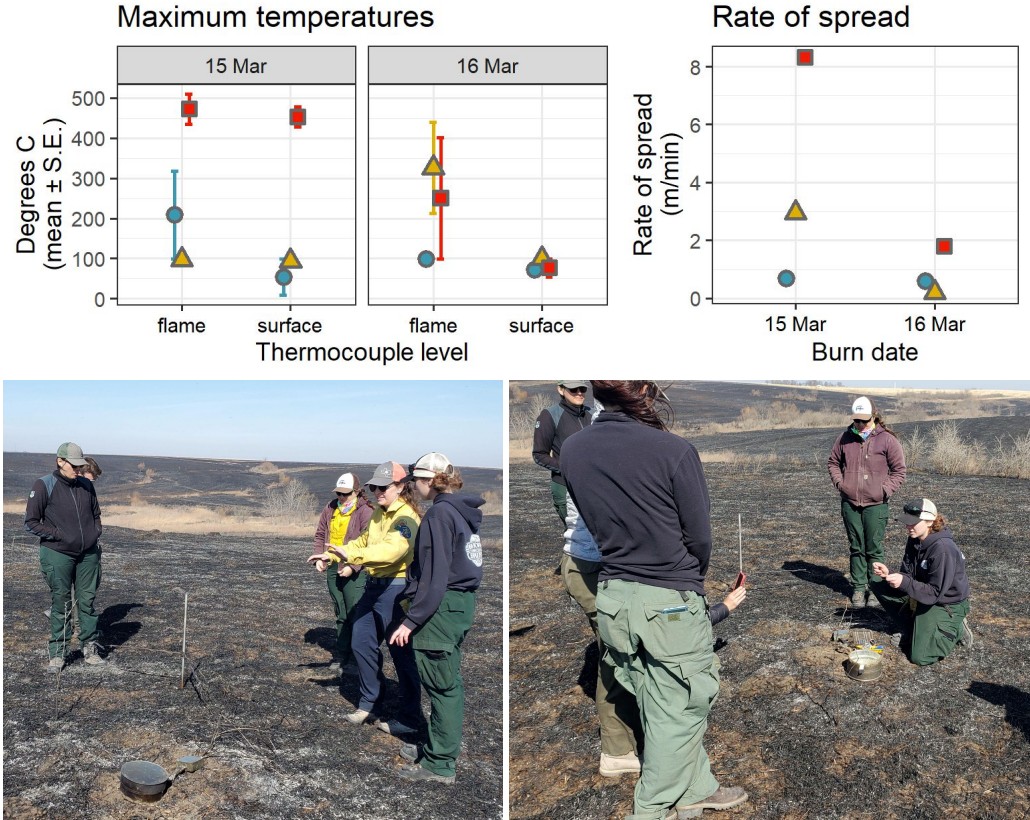

**Figure 8. Top**: Data retrieved from thermocouple dataloggers deployed at three locations in two different burn units at the Dunn Ranch Prairie. Data include flame temperatures measured 15 cm above the soil surface, and soil heating at the soil surface. **Bottom**: After Dr. Wonkka worked with squad leaders to understand how the thermocouple system had performed during the fire event (**L**), squad leaders then briefed the remainder of the crew (**R**). This constituted a contemporary, facilitated application of the "see one, do one, teach one" principle that balances autonomy and supervision.

We used remotely-sensed data to illustrate the variability in burn severity across entire burn units (Figure 9). While not possible to calculate severity from remotely-sensed data during the workshop due to the time necessary for post-burn imagery to be collected and come available, the results of this analysis will be available for educational materials for future workshops.

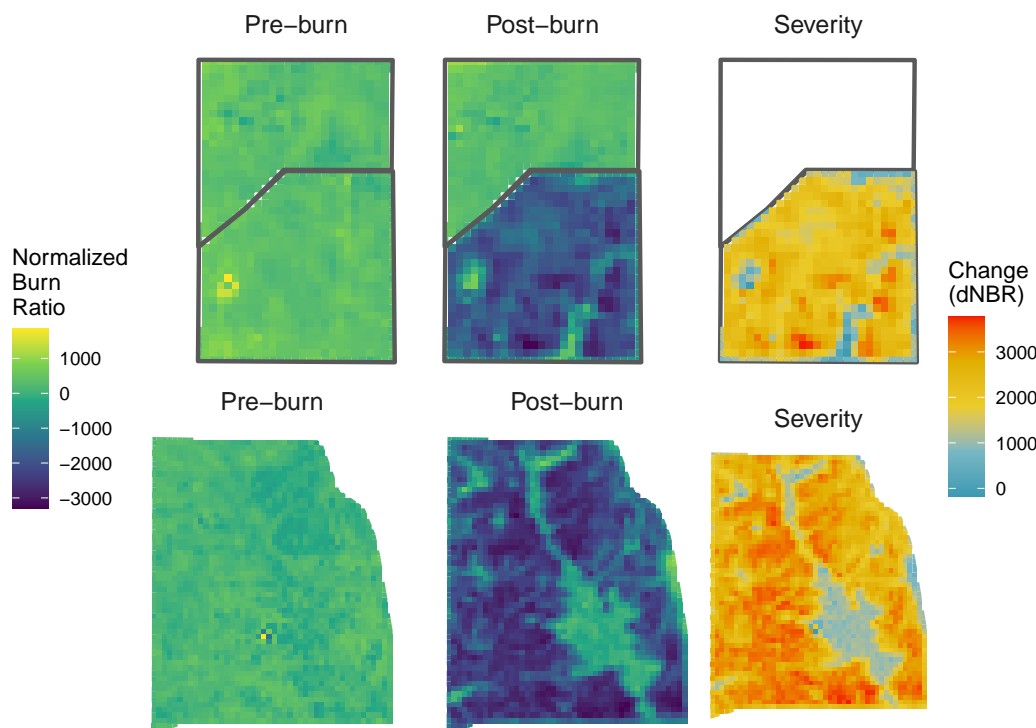

**Figure 9.** A comparison of remotely-sensed data products (LANDSAT 8) from before and after two burns at the Dunn Ranch Prairie. Pre-burn imagery was captured on 1 March 2022, and post-burn imagery was captured on 26 March 2022; fires were conducted on 15 and 16 March 2022 (top and bottom, respectively). The difference in the Normalized Burn Ratio (dNBR) provides a measure of burn severity across each unit (right-most images). Areas with "hotter" colors–increasing from yellow and orange to red–burned with greater severity, i.e., more above-ground plant biomass was consumed.

## 4. After Action Review

To evaluate the success of the workshop in meeting the initial objectives and to identify opportunities to improve, we conducted two After Action Reviews (AAR). First, an anonymous online questionnaire was created and sent to participants, to which 8 of 10 responded. The questionnaire was designed to follow the P.L.O.W.S. format, which participants were introduced to during the course. PLOWS is a structured AAR format designed to focus on five key elements of an operation–Plan, Leadership, Obstacles, Weaknesses, and Strengths–and avoid the erosion of interest in the AAR that can occur when participants state their broad, general opinion upon their initial opportunity to speak (More about PLOWS is available at this online document: https://www.nwcg.gov/sites/default/files/wfldp/docs/plows-presentation.pdf, accessed on 15 August 2022). To more specifically accommodate the evaluation needs for the entire workshop rather than a single incident, we modified PLOWS slightly to operate as PLOWSs, in which "Strengths" is expanded to include "Strengths and Successes". Once the results of the questionnaire were available, the cadre met in a virtual meeting to conduct their own AAR.

Participants appear to have both enjoyed the workshop and gotten substantial value from it, with all respondents either agreeing or agreeing strongly that they would participate again and would recommend the workshop to others (Figure 10). Likewise, the cadre felt that the major objectives were achieved and the workshop went smoothly.

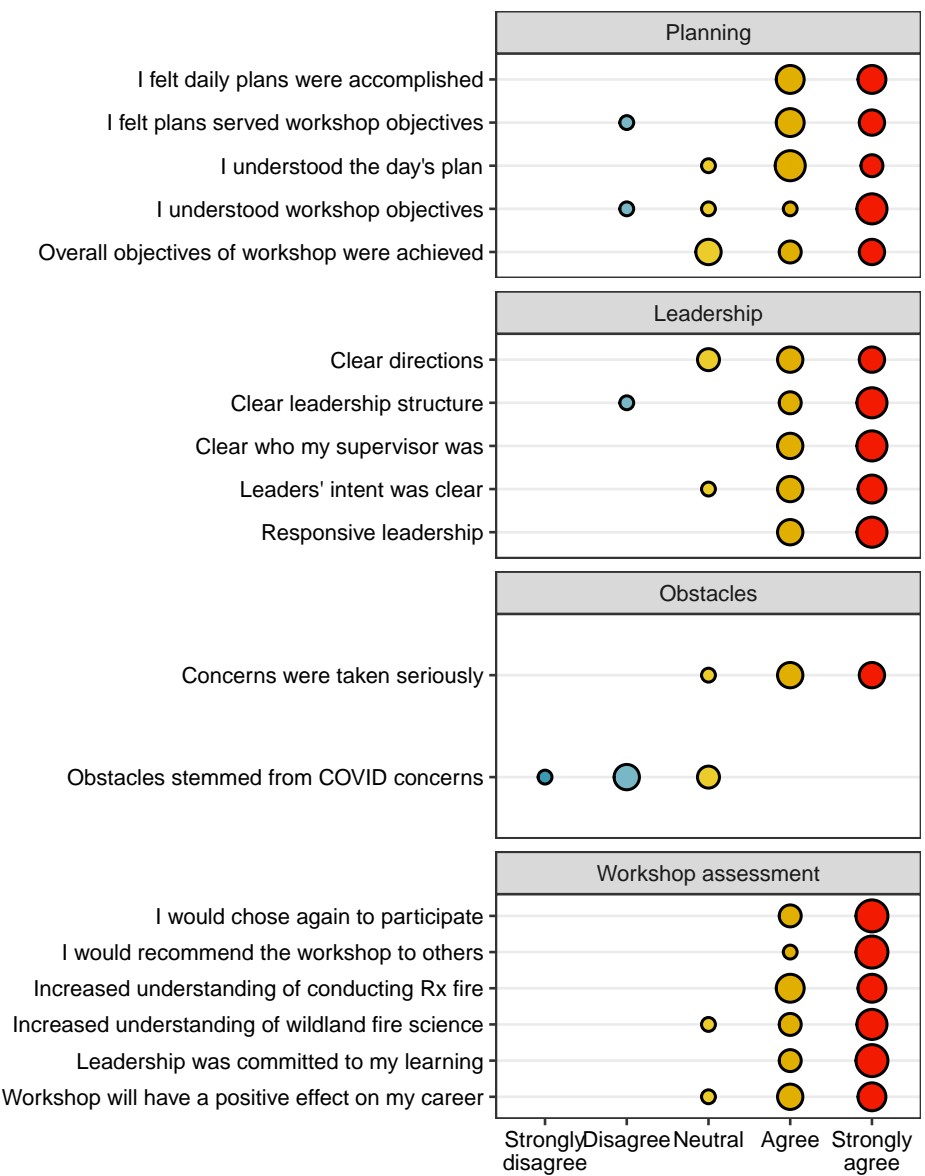

**Figure 10.** Responses from Likert-style questions in the workshop evaluation completed by 8 of 10 participants.

*4.1. Plan*

Once the workshop began, daily plans were made by the cadre the night before and communicated to everyone through daily morning briefings (Figure 11). Participants generally responded favorably when asked how well plans were communicated (Figure 10), although participant responses seemed to indicate that the workshop objectives and the relationship of daily activities to those objectives might be better communicated.

In optional short-answer responses, participants suggested that the objectives of the workshop could have been communicated better ahead of time. Others also suggested that the need for flexibility, especially in terms of adjusting daily plans around weather, be better communicated to participants ahead of arrival. Others suggested covering some of the pre-burn preparation work and planning in the workshop.

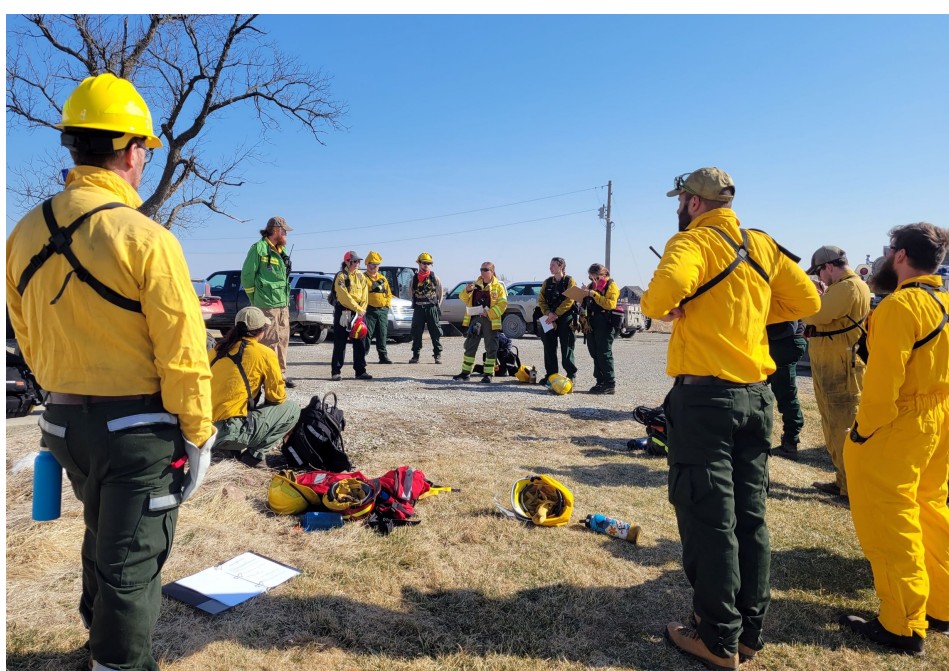

**Figure 11.** The cadre organized daily briefings to communicate the day's activities and objectives to workshop participants.

The suggestion for better communication regarding the extent of flexibility built into the course to adjust for weather served as a reminder to the cadre that the need for flexibility in a multi-day fire operation will be novel to participants who are less familiar with wildland fire. Generally, this was a component of communication regarding leadership intent that the cadre struggled with finding appropriate balance for. Ultimately, they all felt that given differences in participant familiarity with fire, there is a strong need to be clearer on this point in the future. The participants need to understand that this is part of the fire environment and this could be a very valuable part of their learning experience if the cadre incorporate it better into the learning plan.

In their AAR, the cadre was conflicted about how many logistical behind-the-scenes issues relating to even establishing, let alone communicating, plans for the workshop ought to be presented to participants. In reality, all manner of challenges faced the workshop planners, from broad, national scales over the course of many months, to fine local scales on a daily basis. The planning team had already pushed the entire event back a year due to COVID-19 restrictions. Whether TNC could commit to hosting the workshop and whether federally-employed members of the cadre could attend remained uncertain until just weeks before the scheduled start date. Additionally, Dunn Ranch Prairie received substantial snowfall just a couple of days prior to the workshop, and the condition of fuels and trails was uncertain. Fortunately, spring solar intensity and warm breezes melted the snow and dried the burn units while participants were undergoing the first day of training, but the cadre was reviewing their list of various non-live fire exercises, compiled in case just such a contingency was necessary. (These modules included fire behavior modeling exercises using Fireline Handbook Appendix B, Rothermel's nomograms [40], and BehavePlus software [41]; and sandtable simulations of prescribed fire operations and leadership.) While there might perhaps be an educational element to keeping participants informed of persistent uncertainty, the cadre weighed the benefits of this insight for some participants against the risk of overwhelmingly confusing most of them. Confusion can result in poor action implementation, or even emergency situations such as fire escape, injury, or property damage. Therefore, the cadre determined that future workshops should include coursework on the level of flexibility required in fire science and operations to prepare participants for encountering this throughout the course, limiting expectations of a completely fixed

and determined schedule without the need for constant communication regarding ever-changing weather conditions that would overload and confuse some participants.

### 4.2. Leadership

Participants generally reported having a good understanding of the leadership structure for the workshop, and especially who their immediate supervisor was during activities (Figure 10). Importantly, participants reported confidence that the cadre was interested in and responsive to participant needs.

Several criticisms of the leadership clearly reflect the growing pains of a first attempt at a novel event. One participant specifically observed that confusion in the fire science portion on the first day was clarified in that day's AAR and was not a problem for the remainder of the course. Another suggested that more specific timeframes be given for fire science activities to ensure participants were able to keep on pace and not risk holding up the next phase of the operation.

In their AAR, the cadre noted the importance of slowing the pace of briefings and asking participants directed questions to assure that intent is effectively conveyed. AAR for the day was conducted after long days in the field, which was not really avoidable but was recognized as less than ideal. Attempting to leave time before dinner for AAR; or if that is not possible, asking more specific questions to foster engagement could improve AAR by providing the cadre with immediate feedback from participants. In addition, the cadre noted that including an additional instructor would likely have alleviated some of the issues the students noted.

### 4.3. Obstacles

Very few participants noted frequent obstacles during the workshop–six of the eight respondents indicated no obstacles applied to them personally, and five of eight respondents indicated not observing obstacles applying to other participants. However, several obstacles were reported by at least one participant (Figure A1). In comments, participants sought better communication about PPE needs ahead of the workshop for those unfamiliar with the prescribed fire kit. Another comment expressed concern that some participants were fatigued and communication was not clear. No comments indicated what personnel issues among the cadre were perceived, and the cadre was unable to identify points of potential concern, although one Weakness response mentioned the "burn boss was not always calm and collected [which] made some crew members anxious." Obstacles experienced by participants appeared to relate more to the workshop itself than inherent concerns about COVID-19.

Generally speaking, the cadre was surprised participants did not identify more obstacles, especially weather. After residual moisture from the snow at Dunn Ranch Prairie passed as the initial concern, high winds replaced it. Predicted gusts exceeded standard TNC prescriptions, although the cadre was able to mitigate wind conditions by delaying ignition on one unit until winds subsided and burning the unit with the lightest fuels on the day with the highest forecast winds. In addition, Dunn Ranch staff assisted the cadre by undertaking some last-minute fuel mitigation treatment and mowing an area of tall grass that presented a challenge with windy weather prior to the burn; the cadre recognized this cooperation from Dunn Ranch staff as invaluable to the success of the workshop. That these factors were considerations was communicated during briefings, but participants either did not recognize them as obstacles or understood the questionnaire to relate more to their personal experiences than to the obstacles facing the cadre and workshop as a whole. During the AAR, the cadre discussed location and determined that this latitude and fuel type provides the best potential for flexibility and the highest likelihood of having favorable burn conditions while everyone is available to participate in the workshop. Given that, the weather challenges faced during the workshop were essentially part of the best case scenario for conducting a fire science workshop in a grassland setting.

*4.4. Weaknesses*

Participants were asked to identify weaknesses for four separate prompts: specific weaknesses in the workshop relating to fire science activities (five responses), prescribed fire operations (one response), and workshop logistics (five responses), and a general prompt for other weaknesses (one response). Overall, the most frequently-mentioned weakness (three responses) was the lack of opportunity to process and analyze collected data. The fire science instructors were aware of this limitation in the format, and have identified a blended learning opportunity *following* the workshop as a means to offer opportunities to work with data. The cadre believe that the necessary laboratory work to process clippings, download thermocouple data, and assess quality of all data preclude a during-workshop module, although now that one workshop has occurred, opportunities to work with previous years' data exist (e.g., Figure 9). Other respondents generally wanted more time to work with scientific instruments, especially alongside instructors, to better understand elements of the workshop that were, as expected, less familiar to participants. Pre-workshop webinars might allow participants to familiarize themselves with instruments and data collection methods, freeing up workshop time for hands-on experience.

The cadre had a positive take-away from weaknesses expressed by participants; the comments suggested that the participants were confident that they could give the cadre anonymous and frank feedback. They felt that kind of feedback is evidence of a solid leadership team. When participants reflect on their entire experience and share the negative as well as the positive, that provides motivation to improve and provide the best possible personal and professional experience; superficial feedback can foster complacency while frank feedback, such as that received from workshop participants, fosters learning and improvement.

*4.5. Strengths and Successes*

Participants were asked to identify strengths and successes in the workshop, following the same categories as above. In terms of fire science, respondents appreciated the quality of the materials and instruction, the experience of the instructors, and the opportunities to use sampling tools and measurement devices hands-on. Two respondents specifically highlighted the time instructors made available for questions and the value of subsequent discussions. In terms of the prescribed fire operations, several respondents acknowledged the success of the burns, which they attributed to experience on-hand; strong, "decisive" leadership and "excellent" line bosses; and good weather (listed in order of frequency mentioned). Two respondents specifically mentioned crew cohesion. In terms of workshop logistics, respondents acknowledged the quality of the facilities and their proximity to the burn units and the availability and quality of food.

The cadre identified several strengths and successes. Firstly, no personnel were injured and no equipment or property was damaged. All deployed fire behavior instruments performed as intended and no scientific materials were lost through their use in the workshop. Secondly, a number of trainees among the participants were able to achieve hands-on experience that qualified as entries for prescribed fire operations in open position task books (Figure 12). All felt that the small units burned on the first day of the Academy where people can rotate positions and experience the challenges of leadership were a great experience. The cadre saw the opportunity to be in the burn boss position; they could see participants realizing that leading a burn was not as easy as they thought it was going to be, an invaluable lesson for those to lead burns at their home institutions. Small units also present a good opportunity for science, but the cadre did not understand the full capabilities of the site until they were there and running operations and were not able to capitalize on this opportunity for this workshop.

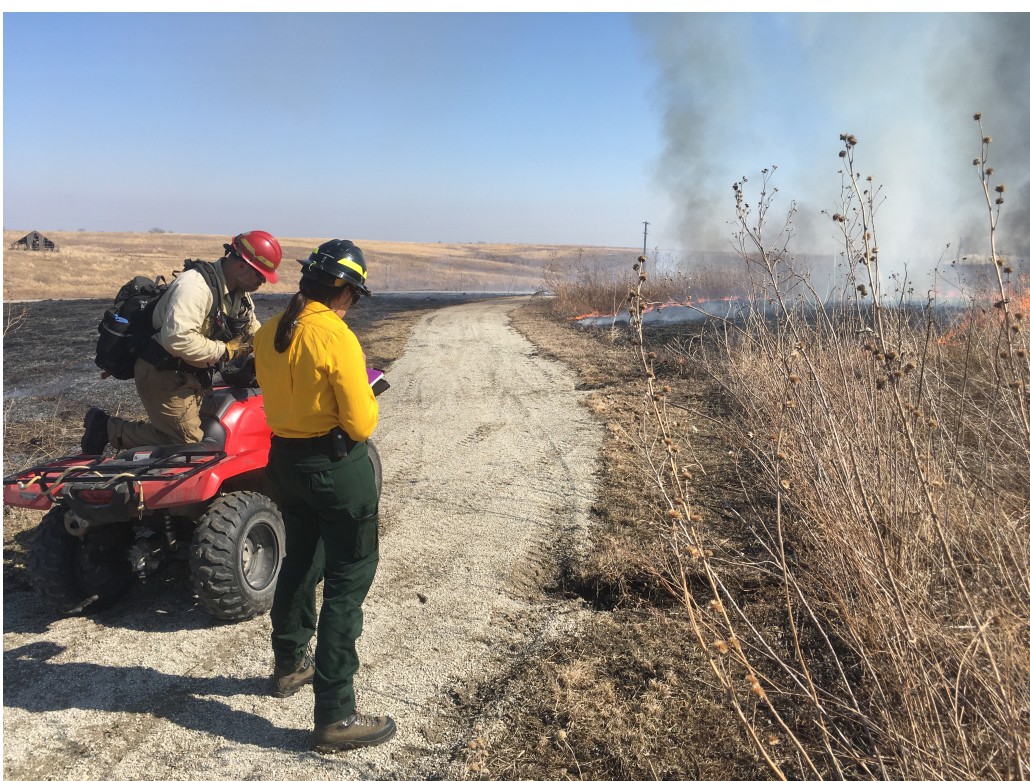

**Figure 12.** A Type 5 Incident Commander (ICT5) trainee who participated in the workshop practices an incident size up on a spot fire that the workshop cadre allowed to slop over across a narrow walking path between two patches of tallgrass fuels used in the first training exercise.

## 5. Discussion

Generally speaking, our hands-on prescribed fire science workshop provided an important opportunity for participants to learn and burn alongside their peers under the supervision of qualified, experienced burners from several agencies, which fills an important training and experience gap among early career wildland fire professionals [14]. However, more specifically, we believe the workshop provided a unique opportunity for participants to gain experience with fire science. While perhaps not at the top of every wildland fire professional's priority skill set, understanding how prescribed fires are conducted and how data are collected are essential components of wildland fire literacy.

The learning environment of the workshop provided an opportunity for students of fire—particularly those relatively new to burning—to experience and learn some of the realities of fire science and operations, including the need for flexibility, the vagaries of weather conditions, and stress inherent in conducting fire operations. In the typical setting of conducting a prescribed burn in a professional capacity or containing a wildfire, there is little time to slow down, ask questions, and fully digest the experience. That is a unique experience that workshops such as this can provide. By participating in a workshop like this, participants are better situated for safety and success in their own wildland fire efforts outside of Dunn Ranch Academy.

Our approach to providing hands-on experience with fire science through training and application resonates with current literature on the efficacy of experience-based learning. For instance, issues with insufficient experience identified in the wildland fire community (Figure 3) arise in medical fields as well, where the long-standing paradigm that surgeons "see one, do one, teach one" to develop operating room skills has come under fire [42]. Hashimoto et al. [43] argue that residents ought to "see more, do more, teach more" to achieve better confidence in their autonomy as independent surgeons. The learning process can be facilitated by giving instruction in alternative formats, such as online training, and offering additional opportunities to gain experience through simulations [42]. George

et al. [44] also describe a four-step model in which instruction begins with "show and tell" and progresses through "active help", "passive help", and finally "supervision only." This echoes the hands-on opportunities we provided for students to watch demonstrations, learn collectively within their groups, and teach their peers under the scrutiny of experts in the field (Figure 8). While the "see one, do one, teach one" approach is still an essential way to acquire new skills outside of classroom environments [45], trust is essential to ensure trainees are honest about their skill levels before teaching others. This role of trust in learning echoes the bonds of empathy [27] we sought to develop with our attempts to facilitate crew cohesion within groups.

While our educational component could be better developed through instructor-moderated coursework ahead of the workshop, blended learning is a double-edged sword in the wildland fire community. On one hand, more individuals have the opportunity to meet training requirements and participate in fire crews, which can help mitigate staffing shortages. On the other hand, quickly working through certification coursework online without sufficient in-person experience can contribute to lopsided professional development (Figure 3) and crews that are perhaps only on paper ready for an incident. However, from the fire science standpoint specifically, our experience and feedback from the AAR suggest there is ample opportunity to incorporate blended learning into several components of this or other workshops, including pre-work and post-workshop modules to work through the data collected. One course participant expressed an interest in an opportunity for workshop participants to deliver presentations on their own professional experience, which would likely be better suited to pre-workshop meetings and would also contribute to participants getting to know each other more prior to arrival on-site.

There are two particular advantages to blended learning that workshops such as ours can leverage [46]. Firstly, doing online coursework prior to the workshop can smooth out imbalances in knowledge and experience among participants. Self-paced material allows those familiar with certain concepts or tasks to move through more quickly while those less familiar with the material can slow down and explore deeper. Secondly, having reviewed material beforehand puts participants in a position to engage instructors at a higher level when interacting face-to-face–with the basic delivery of information largely complete, interactions can more readily focus on synthesis and application. For our workshop, it is likely that students will both gain more education out of blended learning modules as well as enjoy better crew cohesion and get more value from their training and hands-on experience.

## 6. Conclusions and Recommendations

Given the constraints and uncertainties leading up to this workshop, especially as all stakeholders and participants navigated the initial phases of (what we hope turns out to be) the COVID-19 pandemic endgame, we conclude that the Hands-on Prescribed Fire Methods Workshop was a resounding success. Our specific recommendations for future iterations of this workshop, and for anyone planning their own, include:

- *Emphasize expectations in recruitment materials.* Announcements and application materials ought to stress the importance of flexibility and prepare participants for dynamic, often day-of, planning and adaptation.
- *Leverage blended learning.* Use online pre-course work to smooth disparities in prior knowledge of fire science and management among participants. These activities can also facilitate introduction and crew cohesion prior to arrival on-site.
- *Facilitate peer-to-peer learning opportunities.* "See one, do one, teach one" is not a process to be left to its own devices. Facilitators should actively develop crew cohesion and the bonds of empathy among participants that build the trust necessary for group-level problem solving and higher-level learning through co-instruction.
- *Over-emphasize communication.* Briefings, informal group check-ins, and After-Action Reviews are not only essential means to keep participants informed, but also provide opportunities for instructors to receive feedback and increase clarity. Always allow

plenty of time for questions at the end of instructional periods. An adequate number of instructors facilitates learning-while-doing, not simply doing, during periods in the field.

- *Plan alternative activities that still advance objectives.* Some obstacles cannot be planned away. Changes as minor as wind direction or as major as storms or snowfall can leave instructors scrambling to adapt to daily activities. Ensure back-up options are ready and instructors are briefed on how to lead them effectively.

**Author Contributions:** D.A.M., C.L.W., C.M., R.G. and C.W. planned and conducted the Workshop. D.A.M. created the initial draft of the paper, which C.L.W. revised following comments from all authors. All authors have read and agreed to the published version of the manuscript.

**Funding:** This research received no external funding.

**Data Availability Statement:** Not applicable.

**Acknowledgments:** Financial support for the Workshop was provided by the Tallgrass Prairie and Oak Savanna Fire Science Consortium, which receives grant support from the Joint Fire Sciences Program. We recognize substantial in-kind contributions of resources and personnel from The Nature Conservancy, US Fish & Wildlife Service, and USDA ARS (LARRL sponsored scientist travel). We appreciate assistance from Lindsey Reinarz, Daniel Simpson, Kent Wamsley, Missy, and staff at Dunn Ranch Prairie.

**Conflicts of Interest:** The authors declare no conflict of interest.

## Appendix A

**Table A1.** Courses in wildland fire behavior and fire ecology available through the National Wildfire Coordinating Group (NWCG) current training curriculum. *Hours* column includes hours of instruction, both instructor-led and/or self-directed, as appropriate. [†] Note that both Rx-310 and S-490 are followed by specialized intensive 500-level courses that are required to complete their respective qualification series. All URLs accessed on 30 June 2022.

| Course | Hours | Content/Objectives | Target Audience |
|---|---|---|---|
| S-190, Intro. Wildland Fire Behavior https://www.nwcg.gov/publications/training-courses/s-190 | 7/6–8 | Fuels, weather, topography; Recognize critical fire weather, alignment, and danger risk | All qualified crewmembers |
| S-290, Intermediate Wildland Fire Behavior https://www.nwcg.gov/publications/training-courses/s-290 | 37/15 | Tactical implications of interactions between fuels, weather, topography; Causes of extreme fire behavior | All supervisory positions |
| Rx-310, Introduction To Fire Effects[†] https://www.nwcg.gov/publications/training-courses/rx-310 | 32–36/0 | Understand fire as ecological process; fire regime; first-order fire effects; interactions between fire management and natural resources | Rx fire leadership, Resource Advisors |
| S-390, Intro Wildland Fire Behavior Calculations https://www.nwcg.gov/publications/training-courses/s-390 | 42–44/0 | Use and interpret fire behavior prediction models | Incident commanders, Burn bosses |
| S-490, Advanced Fire Behavior Calculations[†] https://www.nwcg.gov/publications/training-courses/s-490 | 44–47/0 | Use advanced techniques to predict fire behavior, project fire growth | Fire and Fire Behavior Analysts, Burn bosses |

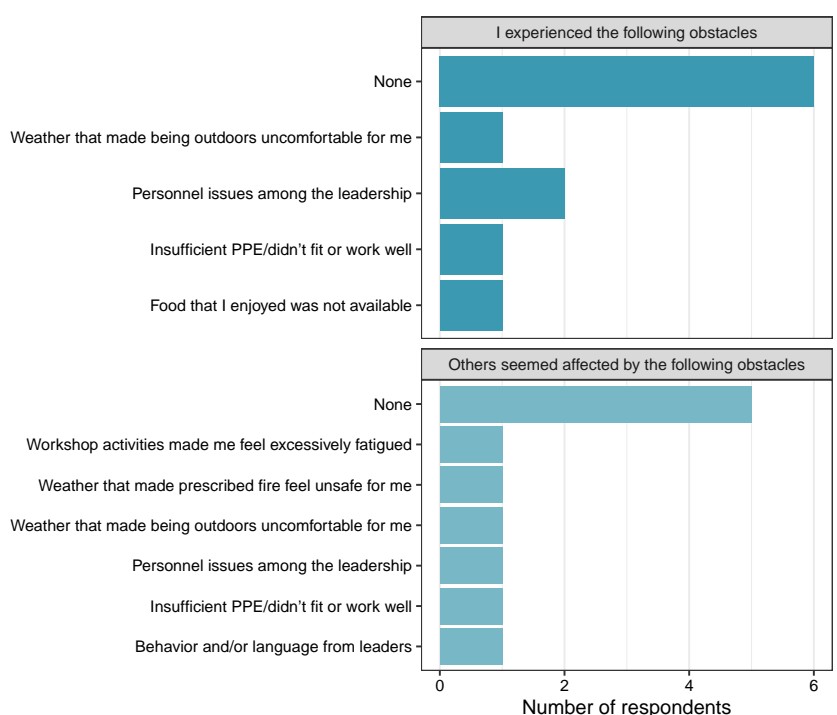

**Figure A1.** Obstacles experienced by workshop participants, or perceived by participants to have affected others.

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
