# Peer review of "The Dunn Ranch Academy: Developing Wildland Fire Literacy through Hands-on Experience with Prescribed Fire Science and Management"

_fire, doi:10.3390/fire5040121_

Round 1

Reviewer 1 Report

This manuscript presents a valuable discussion, but it is ill suited to a scientific journal. I would recommend publication in another outlet such as Fire Learning Network “Notes from the Field” or Wildfire Today. If the authors are intent on publishing in Fire, I would recommend collecting data from multiple workshops with a more rigorous social science framework.

The authors and editors need to improve the clarity on what type of paper this is. The manuscript states it is a ‘Feature Paper’ but based on the Instructions for Authors webpage for Fire, it looks to me like a “Case Report”. Perhaps comments from the other reviewers and me will help make this manuscript ready for publication in Fire, but I believe it is better suited to a different outlet.

Depending on the views of the editor and other reviewers, one option might be to significantly shortening the introduction. Paring down the introduction could remove some areas that are really beyond the scope of the paper and would give more focus to the case study. It is the Dunn Ranch case study that is the real importance of the paper. The conclusion is based the authors opinions and not necessarily transferrable to other workshops.

Overall, the manuscript would benefit from another edit to make the writing more succinct and easier to read. If this manuscript is to be published in Fire, extraneous details from the case study should be excised. For example, though humorous, line 226’s discussion of gravy doesn’t fit in a scientific journal. The goal should be to provide the reader with the key elements that made the workshop successful. For example, it would be better to simply indicate organizers ensured sufficient food rather than providing the details of food logistics.

Specific comments

Line 67 to 69: Given this could be a controversial statement, it would be good to have a citation. I believe some of the papers on barriers to prescribed fire might be appropriate citations (perhaps even Kobziar et al 2009 though expanding to other papers would be good).

Line 113: A paper that might be useful to include in section 1.3:

Hunter, M.E., Colavito, M.M., Wright, V., 2020. The Use of Science in Wildland Fire Management: a Review of Barriers and Facilitators. Curr. For. Reports. 6, 354–367. https://doi.org/10.1007/s40725-020-00127-2

Figure 4. This figure needs further refinement or a better explanation to be useful to the reader. For example, is it suggesting that experience is four times more important than education? Is science less important than management as suggested but the size of the circles?

Line 132 needs a citation

Line 141: TNC’s adaptation of ICS seems more important than just a parenthetical comment. Could more detail and/or a citation be provided?

Line 153: this sentence could be rewritten to be easier to read.

Line 167 to 177 This paragraph could be rewritten to be much more succinct. The same is true for the follow paragraph. For example “As such, the workshop was formally run as an Incident just as any other prescribed fire or wildfire response would be conducted.” Could be simplified to: “The workshop was run as an incident just like any other prescribed fire or wildfire response.” More direct language would make the manuscript easier to read.

Line 180: I’m not sure MDPI’s policy on hyperlinks versus citations, but I’d suggest giving citation along with the hyperlink for Position Taskbooks.

Line 196: TREXs deserve more explanation. While familiar to many readers, others need another sentence or two of description and a citation (e.g. Spencer 2015 which is cited later).

Figure 11 needs to include the number of respondents

Section 4.4 The discussion of weakness is of limited use (and certainly not scientifically valid) because of the small number of respondents.

The conclusion seems entirely based on the authors opinions rather than data.

Reviewer 2 Report

Overall, this is a very well-written paper. But- I am not sure whether it adds to what we know. We already know that learning about fire just in a classroom is not effective. While it makes sense to frame the paper fairly narrowly given the audience, you miss the opportunity to tie to broader literature and knowledge about applied learning and competency based education. This would allow you to place experiences such as blended learning and hands-on burn experience on a scale and reflect about what students learning about fire really need as well as to relate the fire education experience to other applied experiences in ecology etc. What are your specific recommendations?

Reviewer 3 Report

Dear authors,

the article is certainly useful and necessary. It is possible to argue a lot about fires, to write about them, to investigate and count, but if not to teach people not to be afraid of fire, to know it and to increase their practical level of professionalism, all this is useless.

The article is more general educational and social than deeply scientific. In this regard, it is impossible to put it in the structure of IMRAD, to set goals and objectives (although it could have been done there, since pedagogy is also a science), but then the whole article would have lost its identity.

I have not seen such articles in Fire, but in other journals where scientific articles are printed, I have encountered these kinds of reports. If the editorial board considers it possible, such an article can be printed, about the sauce and the buffet should not be written in such detail; correcting Figures 9,11, 12 - empty and too big pictures. Although, if we consider the article as a photo report, then why not.

Round 2

Reviewer 1 Report

Thank you for responding in detail to the reviewer comments. 
This is a difficult paper to review because it is not a standard scientific study and the journal failed to explain it's place in the “Fire Research at the Science–Policy–Practitioner Interface” section. In the context of case studies of community-based fire management, it will be useful.